Network based meta-analysis prediction of microenvironmental relays involved in stemness of human embryonic stem cells

Mournetas Virginie 1 2 vmournet@liv.ac.uk
Nunes Quentin M. 2 3
Murray Patricia A. 1
Sanderson Christopher M. 1
Fernig David G. 2
1 Department of Cellular and Molecular Physiology, Institute of Translational Medicine, University of Liverpool , Liverpool , United Kingdom
2 Department of Biochemistry, Institute of Integrative Biology, University of Liverpool , Liverpool , United Kingdom
3 NIHR Liverpool Pancreas Biomedical Research Unit, Institute of Translational Medicine, University of Liverpool , Liverpool , United Kingdom
Yang Xiang-Jiao
Electronic publication date: 2014 Oct 23
Publication date: 2014
Volume: 2
Electronic Location ID: e618
Received 2014 Jun 10; Accepted 2014 Sep 22
Copyright: © 2014 Mournetas et al.
Copyright year: 2014
Copyright holder: Mournetas et al.
License: This is an open access article distributed under the terms of the Creative Commons Attribution License, which permits unrestricted use, distribution, reproduction and adaptation in any medium and for any purpose provided that it is properly attributed. For attribution, the original author(s), title, publication source (PeerJ) and either DOI or URL of the article must be cited.
License URL: https://creativecommons.org/licenses/by/4.0/

Keywords: Transcriptome, Interactome, Protein–protein interaction network, Human embryonic stem cells, In silico analysis

Funding: North West Cancer Research Cancer and Polio Research Fund National Institute for Health Research Royal College of Surgeons of England-Ethicon Research Fellowship BBSRC This work was partly funded by North West Cancer Research, Cancer and Polio Research Fund, Biomedical Research Unit award from the National Institute for Health Research, Royal College of Surgeons of England-Ethicon Research Fellowship and BBSRC. The funders had no role in study design, data collection and analysis, decision to publish, or preparation of the manuscript.

==============================
Background. Human embryonic stem cells (hESCs) are pluripotent cells derived from the inner cell mass of in vitro fertilised blastocysts, which can either be maintained in an undifferentiated state or committed into lineages under determined culture conditions. These cells offer great potential for regenerative medicine, but at present, little is known about the mechanisms that regulate hESC stemness; in particular, the role of cell–cell and cell-extracellular matrix interactions remain relatively unexplored.

Methods and Results. In this study we have performed an in silico analysis of cell-microenvironment interactions to identify novel proteins that may be responsible for the maintenance of hESC stemness. A hESC transcriptome of 8,934 mRNAs was assembled using a meta-analysis approach combining the analysis of microarrays and the use of databases for annotation. The STRING database was utilised to construct a protein–protein interaction network focused on extracellular and transcription factor components contained within the assembled transcriptome. This interactome was structurally studied and filtered to identify a short list of 92 candidate proteins, which may regulate hESC stemness.

Conclusion. We hypothesise that this list of proteins, either connecting extracellular components with transcriptional networks, or with hub or bottleneck properties, may contain proteins likely to be involved in determining stemness.

Introduction

Human embryonic stem cells (hESCs) are pluripotent cells present in the inner cell mass of the blastocyst (Pera, Reubinoff & Trounson, 2000). They give rise in vivo to the three germ layers (ectoderm, endoderm and mesoderm) and therefore, have the ability to generate all tissues within the body. These cells can also be derived in vitro (Thomson et al., 1998), maintaining an ability to either self-renew or differentiate (Keller, 2005). Human ESCs are a fundamental tool for understanding human embryonic development and constituent mechanisms of differentiation (Keller, 2005). Moreover, they represent a potentially powerful tool in drug screening (Jensen, Hyllner & Bjorquist, 2009) and regenerative medicine (Aznar & Gomez, 2012; Keller, 2005; Wobus & Boheler, 2005). However, in order to mobilise the potential of hESCs, it is necessary to understand the molecular determinants of self-renewal and differentiation.

The core transcriptional network regulating pluripotency (Babaie et al., 2007; Boyer et al., 2005; Chavez et al., 2009; Marson et al., 2008; Rodda et al., 2005) is composed of three transcription factors: octamer-binding protein 4 (OCT4) (Hay et al., 2004), sex determining region Y-box 2 (SOX2) (Fong, Hohenstein & Donovan, 2008) and NANOG (Hyslop et al., 2005; Zaehres et al., 2005). Interestingly, although these transcription factors clearly drive pluripotency (Li et al., 2009; Takahashi et al., 2007), their expression is not restricted to hESCs (Atlasi et al., 2008; Leis et al., 2012; Liedtke et al., 2007; Pierantozzi et al., 2011; Zangrossi et al., 2007). Thus, stemness must in part depend on other hESC specific characteristics, such as the context of expression of these three transcription factors. Protein–protein interaction networks may provide a valuable insight into this hESC specific context (Boyer et al., 2005; Muller et al., 2008). Proteins of the cell microenvironment may also be an important part of this network (Evseenko et al., 2009; Stelling et al., 2013; Sun et al., 2012), since this is the niche where cell–cell and cell-extracellular matrix (ECM) interactions occur, allowing selective cell communication. Indeed, it was through the addition of ECM proteins and growth factors that xeno-free culture conditions for hESCs were defined (Melkoumian et al., 2010; Rodin et al., 2010). These methods have facilitated investigation of the roles that extracellular molecules, such as heparan sulfate (HS) (Stelling et al., 2013), fibroblast growth factor (FGF)-2 (Eiselleova et al., 2009; Greber et al., 2010) and activin A (Xiao, Yuan & Sharkis, 2006) play in hESC self-renewal and differentiation. However, such factors have not always been linked to specific transcriptional networks and many of the defined medium formulations do not completely sustain pluripotency (Baxter et al., 2009; Ludwig et al., 2006). Therefore, other factors involved in the maintenance of stemness must be missing. One key factor could be a wider link between ECM interactions and transcriptional networks, thereby establishing important relay mechanisms between endogenous and exogenous stemness regulators.

Data from large-scale transcriptomic and proteomic studies (Koh et al., 2012) facilitate the construction of large biological networks in which nodes and edges represent molecules and interactions respectively. Studying the topological properties of these networks may enable the elaboration of novel hypotheses. For instance, it has been shown that hubs, which are highly connected nodes within a network, are more likely to be important proteins in a protein–protein interaction network (Jeong et al., 2001), as well as bottlenecks, which are nodes with a high betweenness centrality, meaning many shortest paths within the network pass through them (Yu et al., 2007).

To gain a more global insight into the potential contribution of the cell-microenvironment to stemness, we employed an in silico systems-level approach where a meta-analysis of dozens of microarrays was performed to establish a stringent yet more representative hESC transcriptome. Transcripts of transcriptional and extracellular proteins were used to build a putative interactome or protein–protein interaction network. The organisation of this network was then analysed to identify extracellular proteins with hub or bottleneck properties, which may be involved in determining stemness, as well as proteins connecting the extracellular factors to transcription.

Materials and Methods

Establishing hESC and hESC-derived transcriptomes

The microarray datasets used to establish a high coverage hESC transcriptome were raw data (.CEL image files) of single channel Human Genome U133 Plus 2.0 Affymetrix microarrays downloaded from the ArrayExpress public database (Parkinson et al., 2007). Probe intensity extraction and normalisation procedures were performed with BRB-ArrayTools 4.3.0 beta 1 (Simon et al., 2007) using default median array values (selected by BRB-ArrayTools 4.3.0 beta 1) as reference. The minimum required fold change was 1.5. If less than 20% of the expression values met this value, the gene was excluded. Each individual dataset was first analysed using the three available algorithms: Robust Multi-array Analysis (RMA) (Irizarry et al., 2003), GC-RMA (Wu et al., in press) and Micro Array Suite 5.0 (MAS5.0) (Hubbell, Liu & Mei, 2002). The three lists of expressed genes were either combined to create a total list containing all expressed genes, or compared to create an intersection list containing only overlapping genes. For the hESC datasets, when the intersection list contained at least 50% of the genes of the total list, the dataset was used to perform a meta-analysis to establish the hESC transcriptome. Thus, all hESC datasets matching this criterion were grouped to be analysed together and generate the final intersection list used as the hESC transcriptome for further analysis (Fig. 1). For the hESC-derived cell datasets, if the intersection list contained at least 50% of the genes of the total list, the full transcriptome (fibroblasts and endothelial cells) was used for transcriptomic comparisons; otherwise the datasets were combined to build the final intersection list and form the hESC-derived cell transcriptome, which was used for transcriptomic comparisons (Fig. 1). The identifiers were EntrezGene IDs and Official Gene Symbol identifiers. The identifier conversion was done with the database for annotation, visualization and integrated discovery (DAVID) 6.7 (Huang, Sherman & Lempicki, 2009a; Huang, Sherman & Lempicki, 2009b).

Figure 1 Flow chart of the microarray dataset analysis.

This flow chart describes the microarray meta-analysis process ending by the transcriptome establishment of hESC, endothelial cells, fibroblasts and mixed hESC-derived cells.

Selection of extracellular and transcription related sub-transcriptomes

The extracellular (EC) and the transcription factor related (TF) components of the transcriptomes were extracted using the Gene Ontology (GO) database (Ashburner et al., 2000). The terms used were: GO:0005576 (extracellular region) and GO:0009986 (cell surface) for the EC component; GO:0005667 (transcription factor complex), GO:0008134 (transcription factor binding), GO:0000988 (protein binding transcription factor activity) and GO:0001071 (nucleic acid binding transcription factor activity) for the TF component. Genes involved in biological processes (e.g., cell cycle (GO:0007049), cell adhesion (GO:0007155), cell communication (GO:0007154), cell junction (GO:0030054) and cytoskeleton organization (GO:0007010)) were also highlighted.

By using a published list of HS binding proteins (Ori, Wilkinson & Fernig, 2008), the EC component was divided into two distinct groups: genes coding for HS binding proteins and those coding for non-HS binding proteins.

The hESC transcriptome was compared with the three different hESC-derived cell transcriptomes to establish which mRNAs were only expressed in hESC (the specific part) and which ones were expressed in all analysed transcriptomes (the common part).

Construction and analysis of putative interactomes

Putative interactomes were built with the Search Tool for the Retrieval of Interacting Genes/Proteins (STRING) 9.0 database (Szklarczyk et al., 2011) using interaction data from experimental/biochemical experiments and association in curated databases only, which excludes interaction predictions by neighbourhood in the genome, gene fusions, co-occurrence across genomes, co-expression and text-mining (co-mentioned in PubMed abstracts). A stringent interaction confidence of 0.7 was imposed to ensure a higher probability that the predicted links exist (Szklarczyk et al., 2011).

Analysis of network structure

Cytoscape 2.8.0 software (Shannon et al., 2003) and associated plug-ins were used to visualise and analyse protein–protein interaction networks. Randomised networks were created by the RandomNetworks v1.0 plug-in from the real protein–protein interaction networks. Therefore, each random network had the same number of nodes N and edges L as its corresponding real network. Network topological parameters, such as connected components, average degree 〈k〉, degree distribution P(k), average clustering coefficient 〈C〉, clustering coefficient distribution C(k) and characteristic path length 〈l〉, were computed with the NetworkAnalyser plug-in. Statistical analysis was performed using IBM SPSS Statistics 21 software.

Enrichments analysis of interactome components

Kyoto Encyclopedia of Genes and Genomes (KEGG) (Kanehisa & Goto, 2000) pathway and GO Biological Processes term enrichments were processed using DAVID 6.7 (Huang, Sherman & Lempicki, 2009a; Huang, Sherman & Lempicki, 2009b) for the analysis of transcriptome subsets. Terms were recorded when the EASE score was ⩽ 0.1 and considered significantly enriched when the false discovery rate was ⩽ 0.05. Enrichment was calculated through two different ways: the ratio of the ratio of proteins belonging to the term in the analysed list and the ratio of proteins belonging to the term in Homo sapiens, or hESCs.

Selection of candidate proteins

Proteins with a degree k in the top 20% were considered as hubs, while proteins with a betweenness in the top 20% were considered as bottlenecks (Yu et al., 2007). The EC/TF and specific/common interfaces were established from the hESC sub-interactome, constructed with STRING data (edge confidence of 0.7) and containing EC and TF components only. To be part of the EC/TF interface network, an EC node had to be connected to at least one TF node and vice-versa. Similarly, to be part of the specific/common interface, a specific node had to be connected to a least one common node and vice-versa. In this complete (ALL_EC+TF) list of candidate proteins composed of the two interfaces, hubs and bottlenecks, only the EC nodes from the specific and common parts were kept to establish the final (C+S_EC) short list of candidate proteins (Fig. 2). The KEGG pathway and GO Biological Processes term enrichments were processed as previously. Statistical analysis was performed using IBM SPSS Statistics 21 software and presented as mean ± SEM.

Figure 2 Establishment of the list of candidates, a flow chart.

This flow chart describes the candidate choice process from the hESC transcriptome to the final list of 92 proteins.

Results

The hESC transcriptome

To discover new regulators of hESC pluripotency, 24 hESC microarrays were analysed from four different datasets (Table 1). A total of 8,934 genes were found to be expressed, which constitute the high coverage hESC transcriptome (Table S1). To establish hESC specific expression profiles, three different early hESC-derived cell transcriptomes were extracted from analogous fibroblast (5,086 mRNAs), endothelial cells (5,522 mRNAs) or mixed hESC-derived cells (10,730 mRNAs, Table 1 and Table S1).

Table 1 Microarray datasets analysis.

The access number (second column) gives access to the dataset in ArrayExpress database. Cell types and cell lines (first column), main cell culture conditions (third column), publications linked to the dataset (when available, fourth column) and the number of microarrays used per analysis (fifth column) are specified. Expressed gene lists for each algorithm (RMA, GC-RMA and MAS5.0) as the total and intersection lists are presented. Four hESC datasets (E-GEOD-6561, -15148, -18265 and -26672) have been used to build mix1. Six hESC-derived cell datasets (E-GEOD-9196, -9832, -9940, -14897, -19735 and -21668) have been used to build mix2. Datasets in bold represent the final transcriptomes used for further analysis: mix1 for hESCs, E-GEOD-9832 for the fibroblasts, E-GEOD-19735 for the endothelial cells and mix2 for the mixture of hESC-derived cells.

Cell type (Cell line)	Access number	Linked publication	Main cell culture feature	Number of microarrays	RMA	GC-RMA	MAS5.0	Total	Intersection	
hESCs (H14)	E-GEOD-6561	(Baker et al., 2007)	On feeder cells (irradiated MEFs) FGF2 (4 ng/mL) 20% KnockOut SR	4	9,672	8,680	9,822	11,930	7,088	
hESCs (H1, H7,
H9, H13, H14)	E-GEOD-15148	(Yu et al., 2009)	On feeder cells (irradiated MEFs) FGF2 (100 ng/mL) 20% KnockOut SR OR On feeder-free matrigel Conditioned medium	10	8,798	10,554	10,293	12,467	7,395	
hESCs	E-GEOD-18265	/	On feeder cells (inactivated HFFs) FGF2 (10 ng/mL) 20% KnockOut SR	5	9,602	11,325	10,966	13,499	7,791	
hESCs (H1)	E-GEOD-26672	(Hu et al., 2011)	On feeder cells (irradiated MEFs) FGF2 (4 ng/mL) 20% KnockOut SR	5	9,656	11,285	10,843	12,790	8,235	
hESCs	Mix1	/		24	11,001	12,173	11,546	14,043	8,934	
Embryoid bodies	E-GEOD-9196	(Lu et al., 2007)		9	5,142	6,308	6,667	8,935	3,576	
Blast cells		9	4,142	5,866	6,731	8,617	3,175	
Fibroblasts	E-GEOD-9832	(Park et al., 2008)		3	6,471	7,510	8,072	9,795	5,086	
Neural progenitors	E-GEOD-9940	/		12	6,939	8,432	7,309	11,517	3,944	
Embryoid bodies		3	1,356	2,485	5,009	5,812	942	
Hepatic cells	E-GEOD-14897	(Si-Tayeb et al., 2010)		3	1,669	2,659	4,017	4,864	1,299	
Endothelial cells	E-GEOD-19735	/		4	7,166	9,237	8,448	11,085	5,522	
Embryoid bodies		2	704	1,832	1,156	2,079	583	
Mesenchemal progenitors	E-GEOD-21668	(Evseenko et al., 2010)		3	861	1,482	2,543	3,087	638	
Differentiated cells	Mix2	/		48	16,174	14,172	12,134	17,458	10,730	
Notes.

MEFs mouse embryonic fibroblasts

HFFs human foreskin fibroblasts

SR serum replacer

The mRNAs specifically expressed by the hESCs (1,010 mRNAs) and those common to hESCs and hESC-derived cells (1,933 mRNAs) were identified by comparing the hESC trancriptome with the hESC-derived trancriptomes (Fig. 3A). Gene Ontology (GO) annotation database (Ashburner et al., 2000) was then used to identify the hESC transcription factor (TF) related (721 mRNAs) and extracellular (EC) transcripts. In this last set of mRNAs, a distinction between transcripts coding for HS binding proteins (191 mRNAs) and non-binding proteins (576 mRNAs, Fig. 3B and Table S1) was enabled by a published list of HS binding proteins (Ori, Wilkinson & Fernig, 2008).

Figure 3 Overlaps of transcriptomes and sub-transcriptomes.

(A) Main overlaps of hESC and hESC-derived cell transcriptomes. Grey, hESC transcriptome; Blue, endothelial cell transcriptome; Red: fibroblast transcriptome; Green: mixture of hESC-derived cell transcriptome. (B) The overlaps of hESC sub-transcriptomes. The hESC transcriptome is composed of 8,934 mRNAs in total with a hESC-specific part (1,010 mRNAs, brown part), a common part (1,933 mRNAs, blue part) shared with the hESC-derived cells, and the rest of the mRNAs (grey). Sub-transcriptomes can be highlighted: the HS binding proteins part (191 mRNAs, specific in red and common in pink); the extracellular part (EC) without HS binding proteins (576 mRNAs, specific in orange and common in purple); the transcription factor related part (TF, 721 mRNAs, specific in yellow and common in light blue).

Transcriptome analysis showed that genes known to be involved in stemness were represented in this hESC transcriptome, such as POU class 5 homeobox 1 (POU5F1, which encodes OCT4 protein) (Nichols et al., 1998) and SOX2 (Avilion et al., 2003). As expected, some of these were in the hESC specific subset, such as the telomerase reverse transcriptase (TERT) (Yang et al., 2008) and growth differentiation factor 3 (GDF3) (Levine & Brivanlou, 2006) (Table 2A). Interestingly, NANOG (Chambers et al., 2003) was not present here. Some germ layer markers were also found in the hESC transcriptome, but they were never specific (Table 2B). Lastly, many common additions to cell culture medium, which have been observed to facilitate hESC growth in vitro, such as FGF2 (Eiselleova et al., 2009; Vallier, Alexander & Pedersen, 2005) and activin A (Xiao, Yuan & Sharkis, 2006) were also present (Table 2C).

Table 2 Transcriptomes and literature comparisons, a selection of markers.

‘Transcriptome’ column: transcriptome(s) or sub-transcriptome containing the mRNAs. (A) Signalling molecules required for pluripotency/self-renewal; (B) Germ layer markers and (C) Molecules related to culture medium of hESCs.

	Marker/Family	Acronym	Name	Transcriptome	GO term	
A	Embryonic stem cell	PTEN	phosphatase and tensin homolog	hESC (COMMON)	CS/CA/CC/Cco	
TERT	telomerase reverse transcriptase	hESC (SPECIFIC)		
GDF3	growth differentiation factor 3	hESC (SPECIFIC)	EC non-HS	
NODAL	nodal homolog (mouse)	hESC (SPECIFIC)	CCo/EC non-HS	
ZIC3	Zic family member 3	hESC, Mix	TF	
SOX2	SRY (sex determining region Y)-box 2	hESC, Mix	CC/CCo/TF	
POU5F1	POU class 5 homeobox 1	hESC, Mix	CCo/TF	
B	Ectoderm	NEFH	neurofilament, heavy polypeptide	hESC (COMMON)	CS	
TUBB3	tubulin, beta 3 class III	hESC, Mix	CCo	
Endoderm	KRT19	keratin 19	hESC (COMMON)	CS	
SOX7	SRY (sex determining region Y)-box 7	hESC, Endo, Mix	CCo/TF	
Mesoderm	KDR	kinase insert domain receptor
(a type III receptor tyrosine
kinase) (VEGFR)	hESC, Mix	CA/CCo/HS	
PDGFRA	platelet-derived growth factor
receptor, alpha polypeptide	hESC (COMMON)	CS/CA/CCo	
VIM	vimentin	hESC (COMMON)	CS	
C	Fibronectin	FN1	fibronectin 1	hESC (COMMON)	CA/HS	
ITGA5	integrin, alpha 5 (fibronectin receptor, alpha polypeptide)	hESC, Mix	CA/CCo/J/HS	
ITGB1	integrin, beta 1 (fibronectin
receptor, beta
polypeptide, antigen CD29
includes MDF2, MSK12)	hESC, Endo, Mix	CS/CA/CC/CCo/J/HS	
Fibroblast growth factor	FGF2	fibroblast growth factor 2	hESC (COMMON)	CC/CCo/HS/TF	
FGFR1	fibroblast growth factor receptor 1	hESC (COMMON)	CC/CCo/HS	
FGFR2	fibroblast growth factor receptor 2	hESC, Endo, Mix	CC/CCo/HS	
FGFR3	fibroblast growth factor receptor 3	hESC, Endo, Mix	CCo/J/HS	
FGFR4	fibroblast growth factor receptor 4	hESC (SPECIFIC)	CCo/J/HS	
Activin A	ACVR1B	activin A receptor, type IB (ALK4)	hESC (COMMON)	CC/CCo/EC non-HS	
ACVR1C	activin A receptor, type IC	hESC (SPECIFIC)	CCo	
ACVR2A	activin A receptor, type IIA	hESC, F, Mix	CCo	
ACVR2B	activin A receptor, type IIB	hESC, Endo, Mix	CCo/EC non-HS	
INHBA	inhibin, beta A/Activin A	hESC (COMMON)	CC/CCo/HS	
Notes.

Endo endothelial cell

F fibroblast

Mix mixture of hESC-derived cells

GO term column GO terms found during the GO extraction

CA cell adhesion

CC cell cycle

CCo cell communication

CS cytoskeleton organisation

J cell junction

EC extracellular part

HS heparan sulfate binding proteins

TF transcription factor related part

Putative extracellular/transcriptional interactomes

As the aim of this study was to learn more about the potential importance of functional links between cell/cell–matrix interactions and transcription, putative protein–protein interaction networks containing only transcriptional and extracellular components (EC+TF) were established by means of the STRING database (Szklarczyk et al., 2011) using transcriptional expression data as a proxy for protein expression profiles. Two interactomes were built: one (called ALL) containing all identified EC+TF proteins, composed of 702 nodes and 3,201 edges (Data S1A), and one (called C+S) containing only those transcripts/proteins that were either specific to hESCs or common to hESCs and hESC-derived cells, comprising 209 nodes and 371 edges (Data S1B).

The average clustering coefficient 〈C〉 (indicating the network cohesiveness) was closer to zero for all randomised networks compared to both ALL and C+S interactomes, implying a significantly higher occurrence of clusters in these selected networks (Fig. 4A).

Figure 4 General network parameters of EC+TF putative interactomes.

(A) The average clustering coefficient of real networks and their corresponding average randomised networks with SEM bars (One sample t-test, n = 5, p-value <0.001). (B) The node degree distribution P(k) and (C) the clustering coefficient distribution C(k). C, common part; S, specific part; R, random; EC, extracellular part; TF, transcription factor related part.

As observed in previous protein–protein interaction network studies (Albert, Jeong & Barabasi, 2000; Jeong et al., 2001), both selected networks (ALL and C+S) and randomised networks exhibit a scale free structure, where the degree distribution P(k) follows a power-law P(k) ∼ k−γ, involving the presence of hubs (Fig. 4B and Table S2), and the clustering coefficient distribution C(k) is independent of k meaning there is no inherent presence of modules unlike hierarchical networks, even if there was a tendency of the real networks to be hierarchical (C(k) ∼ k−β) compared to the randomised versions (Fig. 4C).

These results demonstrate that the EC+TF putative protein–protein interaction networks were suitable for further analysis.

Enrichment analysis

GO Biological Processes term and KEGG pathway enrichments were used to determine if the EC+TF putative interactomes contained significantly enriched subsets of proteins. As expected, terms related to EC (extracellular matrix organization), and TF (transcription, DNA templated) appeared. More interestingly, terms relating to development (embryonic development) and pathways already known to be involved in hESC stemness maintenance (transforming growth factor (TGF)- β (James et al., 2005) or wingless-type MMTV integration site family (Wnt) (Sato et al., 2004)) and differentiation (bone morphogenic protein (BMP) signalling (Xu et al., 2005)) were also identified. KEGG Pathways in cancer as well as GO terms of cell differentiation, cell adhesion, cell communication and cell proliferation were represented too (Fig. 5 and Table S3A).

Figure 5 GO/KEGG analyses of EC+TF putative interactomes.

(A) GO Biological Processes term enrichment (against Homo sapiens), in fold change. (B) Percentage of the total number of proteins in Homo sapiens related to GO Biological Processes that are present in ALL and C+S putative interactomes. C, common part; S, specific part.

Fewer terms were found to be significantly enriched when only the common and specific parts (from ALL to C+S) were analysed. However, when they were found significant, the vast majority was more enriched, except the terms related to TF (Table S3A). Nuclear-transcribed mRNA catabolic process (representing 56% of ALL and 48% of C+S) and multicellular organismal development (representing 47% of ALL and 54% of C+S) were the most represented non-related terms (Table S3A).

Interestingly, regulation of cellular component movement was well enriched with fold changes in ALL of 5.8 (Homo sapiens as background)/4.6 (hESC as background) and in C+S of 9.4 (Homo sapiens as background)/7.4 (hESC as background). 28%/51% of the proteins belonging to this term in Homo sapiens/hESC were represented in ALL and 17%/31% in C+S (Tables S3A).

These data show that the EC+TF putative interactomes, both ALL and C+S, still contained the subsets of proteins involved in development, cell differentiation, cell adhesion and cell communication.

Novel proteins potentially associated with stemness

The final list of potential stemness proteins was established from ALL, the EC+TF putative protein–protein interaction network. This list (called ALL_EC+TF) was composed of nodes with hub or bottleneck features, as well as nodes within the specific/common and EC/TF interfaces (Fig. 2). Hubs are thought to be functionally important due to their high number of interactions, while bottlenecks form links between different processes. 58% of the bottlenecks in the ALL network were also hubs. The specific/common interface reflects the links between the more general cell functions and those specific to hESCs. The EC/TF interface represents points of communication between the genome and the cell’s environment, including other cells. The ALL_EC+TF contained 387 candidates (49% EC and 55% TF) with 29 specific (8%) and 126 common (33%) nodes. The key transcription factors OCT4 and SOX2 were present as hubs and part of the EC/TF interface (Table S4A).

Considering GO set (TF, EC) enrichment with regards to proteins belonging to common or specific parts of the transcriptome, the specific subset was enriched in non-HS binding EC proteins (1.7-fold change), whereas the common subset was enriched in HS binding proteins (1.6-fold change) (Fig. 6A). In addition the common subset was found to be enriched in both hubs (1.3-fold change) and bottlenecks (1.6-fold change) (Fig. 6B). Finally, hubs were enriched in TF (1.4-fold change) and bottlenecks in HS binding proteins (1.3-fold change, Fig. 6B).

Figure 6 Comparative enrichment trends within the candidate protein list.

(A) Enrichments in specific, common and other parts with HS, EC non-HS and TF. (B) Enrichments in HS, EC non-HS and TF parts with EC/TF interface, C/S interface, hubs and bottlenecks. C, common part; S, specific part; EC, extracellular part; HS, heparan sulfate binding proteins; TF, transcription factor related part.

To assess the validity of the candidate prediction, a random ALL_EC+TF list was established the same way using a randomised version of the EC+TF putative interactome (Tables S4B). The hub subset was identical in both real and random versions of the candidate list due to the way the randomised network was generated. However, the bottleneck subset in the real list had proteins with significantly higher betweenness centrality (7,456 ± 835, paired sample test, n = 134, p-value <0.001) than the one in the random list (0.0108 ± 0.0005). Moreover, the random list with 581 proteins retained 83% of the original EC+TF putative interactome against 55% for the real list. The comparison between the real list of candidates and its random version showed that the filtering process was meaningful.

Three shortened lists were generated from ALL_EC+TF list to decrease the number of candidates by either keeping only EC proteins (ALL_EC, 188 proteins, Table S4C) or/and C+S proteins (C+S_EC+TF, 155 proteins, Table S4D and C+S_EC, 92 proteins, Table S4E) as described in Fig. 2. 59% of the common proteins in the longest ALL_EC+TF list and 62% of the specific ones were conserved in the shortest C+S_EC list. Similarly, 9% of the hubs and 20% of the bottlenecks were kept.

To determine if each list and each subset (hubs, bottlenecks and interfaces, as well as specific, common and other proteins from the complete (ALL_EC+TF) to the shortest (C+S_EC) list) still contained proteins potentially involved in stemness maintenance, we undertook further GO Biological Processes term and KEGG pathway enrichments (Table S3B–I). Only the subset containing the hESC-specific proteins was found without any significant enrichment regarding the analysed terms and pathways (Table S3C). However, the most represented term in both specific subsets from the ALL_EC+TF and ALL_EC, as well as in the four full lists and in all other subsets, was multicellular organismal development (Table S3B–I).

Again, terms and pathways related to TF appeared in ALL_EC+TF and C+S_EC+TF lists, as well as in all the other subsets of these two lists (Table S3B,D–I). These TF terms and pathways were logically lost in the ALL_EC and C+S_EC lists and subsets.

GO terms related to cell differentiation, cell adhesion, cell communication, cell movement or cell proliferation, and KEGG pathways of cancer were still significantly enriched in the four lists and in the vast majority of the analysed subsets (Table S3B,D–I).

These data demonstrate that the four lists of candidates, as well as each subset of proteins (hubs, bottlenecks, specific/common and EC/TF interfaces) incorporated proteins involved in development and cell communication. Focusing on the EC proteins that were either specific to hESCs or common to hESCs and hESC-derived cells allowed us to reduce the number of candidates to 92 proteins (Table 3 and Table S4E), while insuring that proteins potentially involved in stemness maintenance were retained. Among these proteins, some are already known to be required for maintenance of hESC stemness, either directly, such as NODAL (James et al., 2005; Vallier, Alexander & Pedersen, 2005), FGF2 (Eiselleova et al., 2009; Vallier, Alexander & Pedersen, 2005) and activin A (Xiao, Yuan & Sharkis, 2006), or indirectly through signalling pathways such as TGF-β (James et al., 2005) or Wnt (Sato et al., 2004). Other proteins are also known to play a role in mouse ESC pluripotency, but not yet in hESC, such as the transcription factor 3 (TCF3) (Cole et al., 2008). However, for the majority of candidates, including titin, nothing is known yet about their functions in the context of hESCs.

Table 3 The list of candidates, an overview.

The ‘hubs’ column gives the degree; the ‘bottlenecks’ column gives the betweenness; the ‘S/C’ column indicates if the protein is in the specific/common interface; the ‘EC/TF’ column indicates if the protein is in the EC/TF interface; the ‘KEGG pathway’ column indicates the number of pathway involving each protein (see Table S4 for a complete list).

	Acronym	Name	GO term	Hubs	Bottlenecks	S/C	EC/TF	KEGG pathway	
C	ACTN4	actinin, alpha 4	EC non-HS/CS	17				4	
ACVR1B	activin A receptor, type IB (ALK4)	EC non-HS/CC/CCo			X	X	1	
ADM	adrenomedullin	EC non-HS/CCo		4050.8	X	X	0	
BMP2	bone morphogenetic protein 2	HS/CC/CCo		2693.9	X	X	3	
DMD	dystrophin	EC non-HS/CS/CCo			X		0	
FGFR1	fibroblast growth factor receptor 1	HS/CC/CCo		2634.2	X	X	5	
FN1	fibronectin 1	HS/CA	26	6988.8		X	5	
IL6	Interleukin 6	HS/CCo				X	4	
INHBA	inhibin, beta A/Activin A	HS/CC/CCo			X		2	
ITGA6	integrin, alpha 6 (CD49f)	EC non-HS/CA/CCo/J	40	15218.1		X	6	
ITGAV	integrin, alpha V (vitronectin receptor)	HS/CA/CCo	40	12409.4		X	6	
JAM3	junctional adhesion molecule 3	EC non-HS/CA/CCo/J			X	X	2	
LAMA1	laminin, alpha 1	HS/CA/CCo	15			X	4	
MET	hepatocyte growth factor receptor	HS/CS/CC/CCo	24	10501.7	X	X	6	
PLAT	plasminogen activator, tissue	HS/CCo		2552.8		X	0	
PLAU	plasminogen activator, urokinase	HS/CA/CCo				X	1	
SERPINE1	serpin peptidase inhibitor, clade E, member 1	HS/CA/CCo	24	10022.8		X	1	
SERPINI1	serpin peptidase inhibitor, clade I, member 1	EC non-HS/CA				X	0	
TGFB2	transforming growth factor, beta 2	HS/CA/CC/CCo	25	10580.8	X	X	6	
THBS1	thrombospondin 1	HS/CA/CC/CCo		2066.0		X	5	
VEGFA	vascular endothelial growth factor A	HS/CA/CCo	16	6817.1		X	6	
S	CDH8	cadherin 8, type 2	HS/CA			X	X	0	
FGF4	fibroblast growth factor 4	HS/CA/CCo			X	X	3	
FGFR4	fibroblast growth factor receptor 4	HS/CCo			X	X	4	
IDE	insulin-degrading enzyme	EC non-HS/CCo			X		0	
INHBE	inhibin, beta E	EC non-HS			X		2	
NODAL	nodal homolog (mouse)	EC non-HS/CCo			X		1	
PLXNB1	plexin B1	EC non-HS/CCo			X		0	
TTN	titin	EC non-HS/CS/CC			X	X	0	
WIF1	WNT inhibitory factor 1	EC non-HS/CCo			X	X	1	
Notes.

GO Gene Onlotogy

KEGG Kyoto Encyclopedia of Genes and Genomes

C common part

S specific part

EC extracellular part

TF transcription factor related part

J cell junction

Discussion

We provide a novel picture of the hESC transcriptome built from a meta-analysis and allowing the in silico analysis of a putative hESC protein–protein interaction network. This systems-level approach has been used to identify proteins potentially involved in the maintenance of stemness.

Transcriptomic data provide the most comprehensive insight into variations in cell type or condition specific gene expression profiles. Therefore, data from multiple microarray studies were chosen to generate putative interactomes due to the lack of corresponding comprehensive proteomic profiles. Even if mRNA and protein levels have been suggested to correlate weakly, this correlation may be stronger than anticipated, though this depends on the techniques used to measure mRNA (Li, Bickel & Biggin, 2014; Pascal et al., 2008; Schwanhäusser et al., 2011; Schwanhäusser et al., 2013). Thus, the present study provides a predictive qualitative insight into sub-networks of proteins, which may mediate or maintain human stem cell pluripotency.

The decision to selectively include genes only found by three different algorithms allowed a reduction in the number of false positives in the whole transcriptome, but probably amplified the number of false negatives, which may explain the absence of NANOG. Regarding the specific/common distinction, this pipeline permitted confidence about the common mRNA subset, whereas it likely increased the false positive rate in the specific mRNA subset, which is still half of the common one. However, the use of transcriptomic data from different hESC lines cultured under different conditions highlighted the core transcriptome of these cells.

Not all mRNAs were represented in the putative protein–protein interaction network, probably because coverage of human protein–protein interactions in all databases, including STRING, remains incomplete (De Las Rivas & Fontanillo, 2010). High edge stringency limits imposed in this study should minimise inclusion of false positive interactions (De Las Rivas & Fontanillo, 2010), thereby increasing confidence in the relevance and utility of predicted networks.

The scale-free nature of the EC+TF putative interactomes mean that they should exhibit a high error tolerance thanks to redundancy and a high attack vulnerability, due to the presence of hubs (Albert, Jeong & Barabasi, 2000).

Even incomplete interactomes are very complex structures. In order to focus on the likely most important proteins within this interactome, four selection criteria were applied, the first being the selection of hESC hubs. These proteins constitute a small, but often essential part of the interactome (Awan et al., 2007). For example, deletion of just one hub in yeast is often lethal (Jeong et al., 2001). The second was the selection of bottlenecks, which link processes and so permit cross-talk. The third and the fourth criteria involved were that proteins had to be in the specific/common or EC/TF interfaces. These interfaces are posited to be important, as they reflect communication links between the nucleus and the extracellular matrix, and between the specific and common proteins, which ultimately make hESCs different from other cell types.

Interestingly, the GO term related to cell motility regulation was strongly represented in the candidate lists. Cell movement is a key component of morphogenesis. It is usually accomplished by three steps (protrusion, adhesion and de-adhesion) where cytoskeleton and ECM are involved (Ananthakrishnan & Ehrlicher, 2007). This may be significant as recent data indicate that cell motion may be an intrinsic feature of hESCs (Li et al., 2010).

Regulation of cell proliferation also appeared in our analysis of candidate lists. This may be significant, as cell proliferation is a key property of hESCs, since these cells are able to proliferate almost indefinitely in vitro (Miura, Mattson & Rao, 2004). This capability is sustained by the EC part with growth factors (Activin A (Baxter et al., 2009) and FGF2 (Xu et al., 2005)) and ECM molecules (fibronectin (Baxter et al., 2009) or laminin (Rodin et al., 2010)), as well as by the TF part through the Smad signalling pathway (James et al., 2005; Vallier, Alexander & Pedersen, 2005). Cell proliferation can also be linked to the significant enrichment of cancer pathways in hESCs. Several links arise between cancer and hESCs, for example, the formation of teratomas as a test to assess pluripotency.

Conclusion

Mechanisms involved in stemness are complex, multi-level and determined by the intrinsic cell potential, cell/cell and cell/matrix interactions. The meta-analysis of transcriptomic data in this study has allowed the construction of a hESC putative protein–protein interaction network from which novel ECM proteins have been identified as potential stemness regulators.

Networks are a snapshot of a dynamic model (Assmus et al., 2006; Peltier & Schaffer, 2010). Notions of attractors (or cell stable stationary states), landscapes formed with valleys (attractors) and hills (barriers between attractors), and cell state transitions described by dynamic systems theory will complete this systems biology approach and bring new hypotheses on hESC behaviour (MacArthur, Ma’ayan & Lemischka, 2008; MacArthur, Ma’ayan & Lemischka, 2009; Peltier & Schaffer, 2010; Roeder & Radtke, 2009).

Supplemental Information

Data S1 The hESC putative interactomes

The hESC EC+TF putative interactomes (A) ALL and (B) C+S (C, common part; S, specific part). (Cytoscape file)

Click here for additional data file.

Table S1 List of mRNAs forming hESC and hESC-derived transcriptomes

Each mRNA is identified by the EntrezGene ID, the Official Gene Symbol and the full gene name. ‘Transcriptome’ column: transcriptome(s) or sub-transcriptome containing the mRNAs (hESC: human embryonic stem cells; Endo, endothelial cell; F, fibroblast; Mix, mixture of hESC-derived cells). Colour code (see Legend sheet) indicates different transcriptome and sub-transcriptomes. GO term column, GO terms found during the GO extraction (CA, cell adhesion; CC, cell cycle; CCo, cell communication; CS, cytoskeleton organisation; J, cell junction; EC, extracellular part; HS, heparan sulfate binding proteins; TF, transcription factor related part). (Excel file)

Click here for additional data file.

Table S2 General network parameters

ALL R is the randomised network from the ALL interactome while C+S R is the randomised network from the C+S interactome (C: common part; S: specific part; R: random). Power law of the degree distribution: with R2, the polynomial regression coefficient (degree 2).

Click here for additional data file.

Table S3 GO/KEGG analyses

GO Biological Processes term and KEGG pathway enrichments analysis of (A) EC+TF interactomes, (B) full lists of candidates and (C–I) subsets of candidates (EC, extracellular part; TF, transcription factor related part).

Click here for additional data file.

Table S4 The complete list of candidates

Each protein is identified by its corresponding EntrezGene ID, gene acronym and full gene name. ‘Transcriptome’ column: transcriptome(s) or sub-transcriptome containing the corresponding mRNA (Endo: endothelial cell; F, fibroblast; Mix, mixture of hESC-derived cells). A colour code is applied to aid recognition of each transcriptome and sub-transcriptome (see the Legend sheet). GO term column: GO terms found during the GO term analysis (CA, cell adhesion; CC, cell cycle; CCo, cell communication; CS, cytoskeleton; J, cell junction; EC, extracellular part; HS, heparan sulfate binding proteins; TF, transcription factor related part). “Hubs” column: indicates the number of edges associated with each hub protein. ‘S/C’ and ‘EC/TF’ columns: indicates if a protein is in the specific/common interface or extracellular/transcription factor interface respectively. (A) the longest ALL_EC+TF list of candidates, (B) the random list, (C) the ALL_EC list, (D) the C+S_EC+TF list and (E) the shortest C+S_EC list of candidate proteins, in which C, common part and S, specific part. (Excel file)

Click here for additional data file.

Analyses were performed using BRB-ArrayTools developed by Dr Richard Simon and BRB-ArrayTools Development Team.

Additional Information and Declarations

Competing Interests

Author Contributions

The authors declare there are no competing interests.

Virginie Mournetas conceived and designed the experiments, performed the experiments, analyzed the data, wrote the paper, prepared figures and/or tables, reviewed drafts of the paper.

Quentin M. Nunes conceived and designed the experiments, contributed reagents/materials/analysis tools, wrote the paper, reviewed drafts of the paper.

Patricia A. Murray, Christopher M. Sanderson and David G. Fernig conceived and designed the experiments, wrote the paper, reviewed drafts of the paper.

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
