# Peer review of "Network based meta-analysis prediction of microenvironmental relays involved in stemness of human embryonic stem cells"

_PeerJ, doi:10.7717/peerj.618_

## Round 0.1 · original submission · Major Revisions

Two reviewers agreed to review your paper and one reviewer (reviewer 1) suggested major revision. The other reviewer (reviewer 2) has not submitted his/her review even after several reminders. To avoid furher delay, I am making a decision now.

I have also gone through the paper and found that it is interesting, but I do think that additional analyses would be necessary, such as those three points suggested to reviewer 1 (see the attached comments). In addition, it would be nice to have some validation of the candidates. Related to GO analysis, can you use the GSEA program for additional evaluation?

·

Basic reporting

see blow

Experimental design

see blow

Validity of the findings

see below

Additional comments

It is interesting to analyze gene expression data of stem cells to mine the potential cell-cell interaction genes. I general the analyzing methods are reasonable, however, I have several questions:

1. the cultural conditions of the cells (used for this study) have not been documented. It is possible that the cultural conditions will influence the expression of the genes which are involved in cell-microenvironment interactions. In addition, could the authors find some 3-D culture gene expression data for stem cells to conduct analysis. In this situation, the niche of the stem cells are known.

2. could authors conduct more network motif analysis so that key network motifs could be identified (see PMID: 21674097, PMID: 18091723).

3. String database in a predicted database. I suggest that authors could combine it with the largest manually curated human signalling database ( see http://www.bri.nrc.ca/wang/) and re-run the analysis.

---

## Round 0.2 · accepted · Accept

Thanks for making the suggested revisions.